# A Pilot Study Exploring the Risk of SARS-CoV-2 Infection Among Employees Handling Healthcare Waste in Selected Healthcare Risk Waste Facilities in Johannesburg, South Africa

**DOI:** 10.3390/ijerph22020243

**Published:** 2025-02-08

**Authors:** Neo M. M. Sehlapelo, Phoka C. Rathebe, Nonhlanhla Tlotleng

**Affiliations:** 1Department of Environmental Health, University of Johannesburg, Johannesburg 2028, South Africa; sehlapeloneo23@gmail.com (N.M.M.S.); prathebe@uj.ac.za (P.C.R.); 2School of Health Systems and Public Health, Faculty of Health Sciences, University of Pretoria, Pretoria 0084, South Africa; 3School of Public Health, Faculty of Health Sciences, University of the Witwatersrand, Johannesburg 2193, South Africa

**Keywords:** occupational risk factors, SARS-CoV-2, waste facility, healthcare waste, medical waste handlers, waste management

## Abstract

The SARS-CoV-2 pandemic has increased healthcare waste (HCW) across the globe, giving rise to new challenges such as illegal dumping of medical waste, and an increased risk to hazardous waste exposure such as blood and body fluids that could cause diseases. The study aimed to determine factors associated with SARS-CoV-2 infection among employees handling medical waste in selected healthcare risk waste (HCRW) facilities in Johannesburg, South Africa. The pilot study followed a cross-sectional design, where self-administered questionnaires were used to collect data on occupation-related risk factors for SARS-CoV-2 among HCW handlers working in waste generation, transportation, and final disposal. A total of 33 participants selected from eight HCRW facilities participated in the study. The analysis showed that 21.9% (*n* = 7) of the HCW handlers who participated in the study reported a positive test result for COVID-19, while 78.1% (*n* = 26) reported a negative test result for COVID-19. The logistic regression analysis showed that repeated handling of HCW (COR: 1.50, 95% CI: 1.00–2.25) and not having sufficient hand washing facilities (COR: 1.13: 95% CI: 1.04–1.24) increased the odds of SARS-CoV-2 infection; however, these factors were not significant as risks for SARS-CoV-2. In the multivariable analysis, not being trained on personal protective equipment (PPE) use (AOR: 1.25, 95% CI: 1.00–1.58) increased the odds of SARS-CoV-2 infection, while having 3-6 years of experience in medical waste handling significantly lowered the odds of occupation-related SARS-CoV-2 infection by 33% (AOR: 0.67, 95% CI: 0.48–0.95). These preliminary findings show that factors such as the accessibility of handwashing facilities, training on the use of PPE, years of work experience, and repeated contact with medical waste may play a role in modifying the odds of SARS-CoV-2 infection among HCW handlers. A study with a larger sample size is needed to comprehensively quantify occupation related risk factors associated with communicable disease infections among HCW handlers.

## 1. Introduction

South Africa is the 15th largest medical or healthcare waste (HCW) producer in the world, with approximately 54,425 metric tonnes of HCW generated daily by medical and other public health facilities [1]. The spike in HCW due to severe acute respiratory syndrome coronavirus 2 (SARS-CoV-2) presented significant risks to the environment and the health of those handling bio-hazardous waste. Healthcare risk waste (HCRW) constitutes at least 15% of HCW [2]. HCRW has distinct sub-classes, namely infectious, sharps, pathological, chemical, radioactive, and radio-therapeutic material, cytotoxic and genotoxic waste, and pharmaceutical waste [3,4,5]. South Africa generated 0.001–0.003 tonnes of HCW daily before the SARS-CoV-2 pandemic; however, this increased to 1578 tonnes/day during the SARS-CoV-2 pandemic, making the country the largest producers of HCW in the African continent [6,7].

Healthcare waste management includes medical waste segregation, waste transportation, treatment, and disposal [4]. Employees managing and handling HCW form part of these intricate processes and thus must be trained on correct classification, proper segregation, and storage of the waste to minimise environmental and health exposures [2,4]. Effective management of waste in countries depends on regulations, organisational policies, occupational health and safety (OHS) legislations, training, and the competency of HCW handlers. Approximately 60% of healthcare facilities in low–middle-income countries such as South Africa, Ethiopia, Botswana, Nigeria, Algeria, Kenya, and Nigeria are classified as unfit to manage medical waste [8,9,10,11,12]. In these regions, challenges with incinerators and engineered sanitary landfills due to under-developed infrastructure, unclear waste handling guidelines in HCRW facilities and the lack of or absence of appropriate PPE (face protection (visor)/eye protection (safety goggles), respiratory protection (masks and respirators), body protection (aprons/protective suits), hand protection (gloves), and foot and leg protection (boots/shoes)) increases exposure to pathogens and harmful chemicals in this workforce [10,11,12,13].

HCRW facilities are located in indoor hospital facilities; the exposure rates of infections increase due to high concentration of pathogens, increasing the probable cause of infection in the indoor environments [14]. Thus, this pilot study was undertaken during the COVID-19 pandemic, from November 2021 to November 2023, to assess factors increasing the risk of SARS-CoV-2 infection in HCW handlers in selected healthcare waste facilities in Johannesburg, South Africa. The study serves as a first step for a larger quantitative study that will (1) assess factors that increase exposure to HCRW for HCW handlers; (2) address challenges of poor waste management in South Africa; and (3) provide recommendations to improve policies in HCW facilities to reduce the cumulative incidence of infection during pandemics.

## 2. Materials and Methods

### 2.1. Study Design and Study Sites

The pilot study followed a cross-sectional study design to collect quantitative data on HCW handlers. The study participants were selected from 8 healthcare risk waste (HCRW) facilities (Sites A to H) as shown in Figure 1, from the 11 HCRW facilities (5 treatments and 6 incinerators) formally registered in Johannesburg, South Africa [15]. The random selection of eight HCRW facilities provided a representative sample of study sites for use in the pilot study.

### 2.2. Study Population and Sampling

HCW handlers working as waste collectors, waste transporters, and waste disposers with more than one-year experience were randomly selected to participate in the study. Study participants were male and female, aged 18–65 years, and were confirmed to have been employed throughout the SARS-CoV-2 epidemic in South Africa. A random selection strategy was used to select study sites, ensuring that all eleven (11) of the facilities in Johannesburg had an equal chance of being included in the study. Study participants were selected based on the number of HCW handlers in the selected facility. In site B, 1 of 2 HCW handlers were selected, while in site H, 12 of 20 HCW handlers were selected to participate in the study. Participants from sites A, C to G were selected using a snowballing sampling strategy, based on the availability of the HCW handlers.

### 2.3. Data Collection (Tools and Methods)

Data were collected using a close-ended structured questionnaire. The questionnaire was designed on Google Forms and was sent to participants who consented to participate in the study. Participants who were unable to complete the survey online were interviewed via telephone, while others were provided with hard copies of questionnaires. The questionnaire was pre-tested on ten HCW handlers selected from one study site to ensure validity and reliability.

The questionnaire was divided into three categories: demographic characteristics; occupation-related factors, risk factors for SARS-CoV-2 infection as reported in the literature, and SARS-CoV-2 vaccination status of participants. Participants were asked about their role as an HCW handler, the type of waste at the selected HCW facility, years of work experience as an HCW handler, training received on handling medical waste, education level, adequate use of PPE, existing comorbidities, and lifestyle factors such as smoking and alcohol use.

### 2.4. Data Management and Analysis

Data cleaning and statistical analysis were conducted using the Statistical Package for Social Sciences (SPSS), version 29.0.0.0 (241), and Statistical Software for Data Sciences (STATA), version 16.1. Categorical variables such as sex, race, level of education, years of experience, and use of PPE were summarised as frequencies and percentages. A chi-squared test/Fischer exact test was used to assess the statistical association between categorical variables. A multivariable logistic regression analysis was used to assess the relationship between demographic variables, work-related factors, and other lifestyle factors and the odds of SARS-CoV-2 infection (coded as “YES” (0) having a positive laboratory test result for SARS-CoV-2 and “NO” (1) for HCW handlers who had never tested positive for SARS-CoV-2 infection). Variables found to not be statistically significant in the univariate regression analysis were not included in the multivariable model during model building. The results were presented as unadjusted/crude odds ratios (CORs) in the univariate analysis for all covariates and adjusted odds ratios (AORs) in the multivariate analysis, with 95% confidence intervals. Variables were reported as statistically significant at a 5% significance level (*p* < 0.05).

## 3. Results

### Characteristics of the Study Population

Table 1 shows the socio-demographic characteristics of the study participants. The majority of the participants (60.6%, *n* = 20) were between 36 and 49 years old, while 27.3% (*n* = 9) were aged 18–35 years, and 12.1% (*n* = 4) were aged 50–65 years. Twenty-eight of the selected participants were male (84.8%, *n* = 28), employed on a full-time basis (93.9%, *n* = 31), and worked in urban facilities (78.8%, *n* = 26), with 18.2% (*n* = 11) having 3–6 years of experience, 33.3% (*n* = 11) with 7–10 years, and 27.3% (*n* = 9) with more than 10 years of experience. Most HCW handlers (90.6%, *n* = 29) reported no comorbidities, and a high percentage (87.9%, *n* = 29) reported having been vaccinated. The cumulative incidence of SARS-CoV-2 among employees handling medical waste in the study population was 22% (cumulative incidence = 7/33 × 100 = 21.2%). Notably, 71.4% (*n* = 5) of SARS-CoV-2-positive cases were among those with 3–6 years of experience, while no cases were reported among those with 0–2 years or more than 10 years of experience.

Table 2 reports the association between various demographic factors and SARS-CoV-2 infection among HCW handlers in selected HCRW facilities. Among the factors analysed, years of experience showed a significant association with SARS-CoV-2 infection (*p* = 0.050). Other demographic factors did not show a significant association with SARS-CoV-2 infection. Education levels varied among positive cases, with 57.1% (*n* = 4) having completed Grade 12 (*p* = 0.190). Most positive cases (85.7%, *n* = 6) were full-time employees (*p* = 0.395). Regarding facility location, 57.1% (*n* = 4) of positive cases were from urban facilities (*p* = 0.110). Comorbidity status and vaccination status did not show significant associations, with 85.7% (*n* = 6) of positive cases having no comorbidities (*p* = 0.550) and being vaccinated (*p* = 0.536).

The cumulative incidence of SARS-CoV-2 among employees handling medical waste in the study population was 22% (cumulative incidence = 7/33 × 100 = 21.2%) as shown in Figure 2. As illustrated in Table 1, 71.4% (n = 5) of SARS-CoV-2-positive cases were among those with 3–6 years of experience, while no cases were reported among those with 0–2 years or more than 10 years of experience.

Table 3 presents a bivariate and multivariable analysis of socio-demographic and occupational factors associated with SARS-CoV-2 infection risk among employees handling medical waste in selected HCRW facilities in Johannesburg, South Africa. In the bivariate analysis, the odds of not having training on PPE use showed a 1.13 times higher chance of contracting SARS-CoV-2 infection (COR: 1.13, 95% CI: 1.04–1.23, *p* = 0.005) compared to having been trained for PPE use. Similarly, the absence of handwashing facilities was linked to increased odds (COR: 1.13, 95% CI: 1.04–1.24, *p* = 0.005) of SARS-CoV-2 infection compared to facilities that contained handwashing facilities. Other factors in the bivariate analysis were not found to have a statistically significant associations with SARS-CoV-2 infection, including all forms of PPE protection (COR: 0.83, 95% CI: 0.52–1.34, *p* = 0.453), vaccination status (COR: 0.93, 95% CI: 0.66–1.30, *p* = 0.669), presence of comorbidities (COR: 1.07, 95% CI: 0.77–1.50, *p* = 0.688), handling all waste forms (COR: 0.82, 95% CI: 0.59–1.16, *p* = 0.265), lack of a booster shot (COR: 1.13, 95% CI: 0.95–1.36, *p* = 0.174), and not wearing gloves and mask (COR: 1.21, 95% CI: 0.86–1.71, *p* = 0.265).

In the multivariable analysis, education level remained a significant factor, with HCW handlers having postgraduate education having 39% decreased odds of SARS-CoV-2 infection (AOR: 0.61, 95% CI: 0.45–0.84, *p* = 0.003) compared to HCW handlers with primary school education. Years of experience also remained significant, with those having 3–6 years of experience having a lower risk (AOR: 0.67, 95% CI: 0.48–0.95, *p* = 0.023) of SARS-CoV-2 infection compared to HCW handlers with less than 2 years of work experience. Lack of PPE training indicated a marginally significant association with increased odds of reporting a SARS-CoV-2 infection (AOR: 1.25, 95% CI: 1.00–1.58, *p* = 0.051) compared to those who were trained in using PPE. HCW handlers in the age categories of 36–49 years (AOR: 1.02, 95% CI: 0.79–1.32, *p* = 0.862) and 50–65 years (AOR: 1.01, 95% CI: 0.83–1.22, *p* = 0.970) did not show a statistical significant difference in association for SARS-CoV-2 infection when compared to HCW handlers in the 18–35 years’ age group. Postgraduate education levels lowered the odds of s SARS-CoV-2 infection in HCW handlers (AOR: 0.61 (0.45–0.84), 95% CI: 0.45–0.84) compared to HCW handlers with primary school education.

## 4. Discussion

In this pilot study, occupational factors increasing the odds of SARS-CoV-2 infection in HCW handlers in selected healthcare waste facilities in Johannesburg were assessed to pave the way for a larger quantitative study on this workforce. Medical waste handlers encounter numerous occupational health risks due to the nature of their work. The preliminary findings indicate that PPE training and handwashing facilities, among other factors, may play a role in the risk of infection in those handling medical waste. The cumulative incidence of SARS-CoV-2 among medical waste handlers in this pilot study was 21.9%. The cumulative incidence of SARS-CoV-2 reported in the Gauteng City Region (GCR) between March 2020 and December 2022 was 33.2% (1,343,026 individuals) [16]. GCR reported the greatest admission rate of HCW across all provinces during the first and second wave of the SARS-CoV-2 pandemic [17].

Number of years of work experience showed a statistically significant association with reporting SARS-CoV-2 infection. This is supported by another study, which reported that the probability of exposure to occupational health risks may either increase or decrease depending on the years of work experience [18]. In this preliminary study, years of experience showed a significant association with infection, with HCW handlers with 3–6 years of experience presenting a lower risk of testing positive for SARS-CoV-2 compared to HCW handlers with 0–2 years of experience. This was similar to reports in the literature that found that the expertise and approach of HCW handlers were linked to years of experience.

SARS-CoV-2 positive cases were reported in the age group 36–49 years (57.1%). The respondents between the ages of 50 and 65 had 1.23 times increased odds of contracting a SARS-CoV-2 infection compared to those in the age group 18–35 years, aligning with Chen and colleagues, which highlights age as a critical factor in the severity and progressions of diseases. Moreover, mortality from SARS-CoV-2 and other acute diseases are linked to ageing as a risk factor [19]. The highest education level assessed was postgraduate, and in the multivariate analysis, this remains a protective factor (AOR: 0.61, 95% CI: 0.45–0.84, *p* = 0.003), especially when compared to primary school education; thus; the higher the level of education, the more compliance with precautionary measures linked with public health education on SARS-CoV-2 is attained [20]. In this study, the education levels of workers who obtained Grade 12 presented a high risk of 1.23, which did not vary from those who obtained a primary school education. In comparison, those who obtained diplomas/degrees had a risk of 1.13. Furthermore, education is important because, in a study conducted in Brazil on urban cleaning and solid waste management workers, a reluctance to vaccinate was observed in individuals with lower education levels [21].

A lack of PPE training showed a marginally significant association with increased risk (AOR: 1.25, CI: 1.00–1.58, *p* = 0.051); thus, in correspondence with other studies which depicted reduced health risks being linked directly with PPE use post-release amidst the challenges that rose during the SARS-CoV-2 pandemic [22,23]. Furthermore, it was observed in a Pikitup© Depot in Roodepoort, Gauteng, that PPE use correlates with SARS-CoV-2 infection when employees tested positive, leading the facility to shutdown [24]. In addition, in a study undertaken at a hospital in the United States, it was found that although the waste workers had a public health education level of a score of 100%, due to the poor distribution of PPE use (8 out of 50 waste workers), more than 90% of them were exposed to illness due to incorrect handling of medical waste [25]. Respiratory illnesses were among some of the illnesses that the waste workers were exposed to, and this accounted for 18 cases (36%) [25]. Thus, while PPE use is important, training on its use is equally important.

Urban facilities demonstrated a higher risk of 1.23, although it was not statistically significant. This high risk is a result of the high influx of medical waste into urban facilities because they were more accessible, which may inherently lead to higher exposure levels compared to township/rural facilities, which are the least accessible.

Working outside of hospitals, having more than 7 years of experience (7–10 years), and having 10 years of experience were not statistically associated with risk of infection. Similarly, in a Ugandan study, it was shown that the probability of exposure to occupational health risks may either increase or decrease due to an individual’s years of experience [18]. Furthermore, according to Dinoi and colleagues, hospitals are indoor environments, and their surfaces present more contamination. Hence, SARS-CoV-2 was observed in 85.3% of the samples from hospitals [3].

In the unadjusted model, the frequency of PPE use showed a strong association, with those always using PPE having a higher risk of infection compared to frequent users (COR: 1.81, 95% CI: 1.67–1.95, *p* < 0.001). Handling all forms of waste is perceived to involve a higher risk; however, the magnitude of risk is linked to the levels of exposure from the type of medical waste handled. Use of gloves and masks shows and increased risk of SARS-CoV-2 when they are not used to handle waste. Nonetheless, due to the small number of observations in these variables, adjusted odds ratios could not be obtained in the final model.

Comorbidities and vaccination presented no significant associations, as 85.7% of the positive cases presented the absence of comorbidities. In this study, 14.3% of participants with comorbidities tested positive for SARS-CoV-2; this result is contrary to a study conducted in Brazil on urban cleaning and solid waste management workers, were a 22.5% prevalence of SARS-CoV-2 infection was reported among urban cleaning and solid waste management workers during transmission of the Omicron variant in Brazil [21]. In the Brazil study, (87.2%) of participants reported no pre-existing comorbidities.

The frequency of medical waste collection in this study, specifically collecting waste four times a week, was not statistically significant for SARS-CoV-2 infection. This may be because of the type of waste that was collected, which was not associated with a high risk of infection. The scientific literature indicates that “vaccine-induced immunity” is a primary defence against infectious diseases, including SARS-CoV-2 [26]. The absence of sufficient handwashing facilities showed a non-significance relationship with reporting SARS-CoV-2 infection. This may be attributed to HCW handlers’ frequent use of PPE and handling waste in designated boxes in accordance with the training they received. This collective effort, therefore, minimises the contamination of surfaces. This is further supported by scientific research conducted which depicted that SARS-CoV-2 infection thrived on contaminated surfaces, similar to exposure to infected droplets through inhalation [27].

Age was not significant in this study; thus, our results do not correspond with the scientific literature which shows that mortality from SARS-CoV-2 and other acute diseases is linked to ageing as a risk factor [14]. According to Chen and colleagues [19], the sex of an individual is linked to certain behavioural practices. In this study, statistical significance was not presented by the variable sex. This may have resulted from the swift application of methodologies from the training the medical waste handlers received at work. Marital status was not significant in this study, thus not corresponding with a study indicating that living with a spouse safeguarded individuals against poor mental health [28]. This may have been because the spousal dynamics were favourable, thus reducing stress.

This pilot study presents a number of limitations, and biases, including selection bias during sampling of study participants and recall bias, where participants may not recall critical information related to factors investigated in this study. Although the cross-sectional study design aided in establishing the magnitude of the association between the study variables, the limitations of its use include its temporal associations and its inability to establish causal factors.

## 5. Conclusions

This pilot study assessed occupational factors that increased the odds of SARS-CoV-2 infection in HCW handlers in selected healthcare waste facilities. This study serves as a first step towards a larger quantitative study that will address occupational health risks affecting HCW handlers to improve education, policies, and legislation for waste management and awareness on infection control in HCRW facilities in South Africa.

## Figures and Tables

**Figure 1 ijerph-22-00243-f001:**
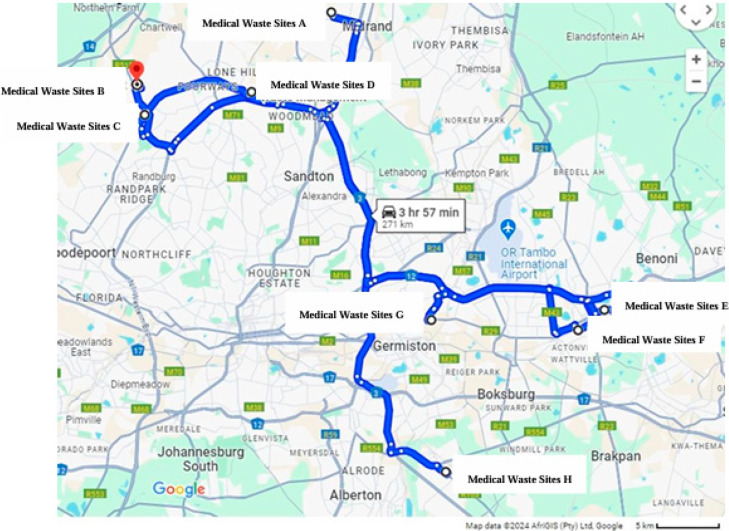
Healthcare facilities selected in the study (*Figure extracted from google maps*).

**Figure 2 ijerph-22-00243-f002:**
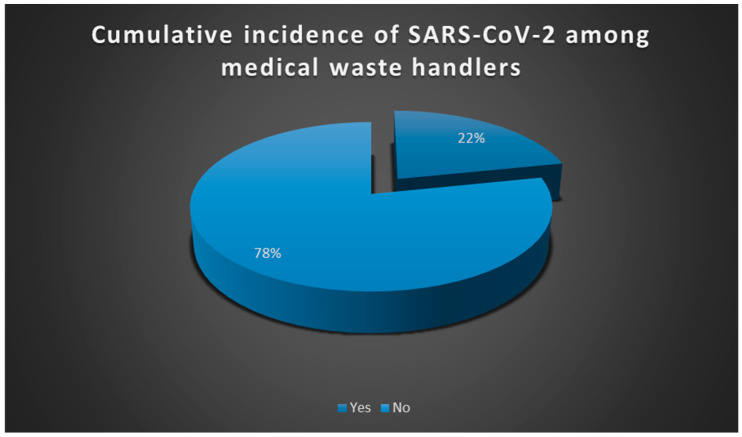
Cumulative incidence of SARS-CoV-2 risk among employees handling medical waste in selected healthcare risk waste (HCRW) facilities in Johannesburg, South Africa.

**Table 1 ijerph-22-00243-t001:** Characteristics of the study population of employees handling medical waste in selected healthcare risk waste (HCRW) facilities in Johannesburg, South Africa (N = 33).

Variable	Category	Frequency	Percentage (%)
Age in Years	18–35	9	27.3
	36–49	20	60.6
	50–65	4	12.1
Sex	Male	28	84.8
	Female	5	15.2
Marital Status	Single	11	33.3
	Living with a partner/married	21	63.6
	Missing	1	3.0
Level of Education	Primary School	6	18.2
	Grade 12	21	63.6
	Diploma or Degree	2	6.1
	Postgraduate	1	3.0
	Missing	3	9.1
Employment Contract	Part-time	2	6.1
	Full-time	31	93.9
	0–2	2	6.1
Years of Experience	3–6	11	33.3
	Yes	1	3.0
Working in a hospital	No	32	97.0
	7–10	11	33.3
	More than 10	9	27.3
Facility Location	Rural	6	18.2
	Urban	26	78.8
	Missing	1	3.0
Use of PPE	Yes	31	93.9
	No	0	0
	Missing	2	6.1
Comorbidity	Yes	3	9.1
	No	29	87.9
	Missing	1	3.0
Vaccination Status	Yes	29	87.9
	No	3	9.1
	Missing	1	3.0
Laboratory confirmed	Positive	7	21.2
SARS-CoV-2 infection result	Negative	25	75.8
	Missing	1	3.0
	HCW generation	1	3.0
Occupation/role	HCW collection	22	66.7
	HCW final disposal	12	36.4

**Table 2 ijerph-22-00243-t002:** Chi-squared/Fischer test of cumulative incidence of SARS-CoV-2 infection and demographic factors among employees handling medical waste in selected healthcare risk waste (HCRW) facilities in Johannesburg, South Africa (N = 33).

Variable	Category	SARS-CoV-2 Infection (Positive/n%)	SARS-CoV-2 Infection (Negative/n%)	*p*-Value
Age in Years	18–35	3 (42.9)	5 (20.0)	0.316
	36–49	4 (57.1)	16 (64.0)	
	50–65	0 (0.0)	4 (16.0)	
Sex	Male	6 (85.7)	21 (84.0)	0.704 **
	Female	1 (14.3)	4 (16.0)	
Marital status	Single	3 (42.9)	8 (32.0)	0.667 **
	Living with a partner/married	4 (57.1)	17 (68.0)	
	Primary School	2 (28.6)	3 (13.6)	0.190
Level of Education	Grade 12	4 (57.1)	17 (77.3)	
	Diploma or Degree	0 (0.0)	2 (9.1)	
	Postgraduate	1 (14.3)	0 (0.0)	
Employment Contract	Part-time	1(14.3)	1 (4.0)	0.395 **
	Full-time	6 (85.7)	24 (96.0)	
Work in Hospital	Yes	0 (0)	1 (100)	0.591
	No	1 (100)	0 (0)	
	0–2	0 (0.0)	2 (8.0)	0.050 *
Years of Experience	3–6	5 (71.4)	5 (20.0)	
	7–10	2 (28.6)	9 (36.0)	
	More than 10	0 (0.0)	9 (36.0)	
Facility location	Rural	3 (42.9)	3 (12.5)	0.110 **
	Urban	4 (57.1)	21 (87.5)	
Use of PPE	Yes	7 (21.9)	25 (78.1)	.^a^
	No	7 (21.9)	25 (78.1)	
PPE training	Yes	7 (21.9)	25 (77.4)	0.591
	No	0 (0)	1 (100)	
Comorbidities	Yes	1 (14.3)	2 (8.3)	0.550
	No	6 (85.7)	22 (91.7)	
Vaccination	Yes	6 (85.7)	23 (92.0)	0.536 **
	No	1 (14.3)	2 (8.0)	
Laboratory confirmed	Yes	0 (0)	7 (100)	0.001 *
SARS-CoV-2 infection results	No	25 (100)	7 (100)	
Occupation/role	HCW generation	7 (22.3)	24 (77.4)	0.591
	HCW collection	5 (22.7)	17 (77.3)	0.863
	HCW final disposal	2 (18.1)	9 (81.8)	0.715

n%: number percentage, * significant, ** no significant association, .^a^ no statistics were computed because PPE use is a constant.

**Table 3 ijerph-22-00243-t003:** Logistic regression analysis assessing the relationship between demographic and work-related factors and SARS-CoV-2 infection among HCW handlers in selected HCRW facilities in Johannesburg, South Africa (N = 33).

Variable	Category	Crude Odds Ratio (COR) (95%CI)	*p* Value	Adjusted Odds Ratio (AOR) (95%CI)	*p*-Value
Age	18–35	ref (1.00)	-	1.00 ref	-
	36–49	1.11 (0.88–1.39)	0.387	1.02	0.862
	50–65	1.23 (1.00–1.52)	0.052 *	1.01	0.970
Sex	Male	1.00 ref	-		
	Female	1.01 (0.81–1.26)	0.911		
Marital status	Single	1.00 ref	-		
	Living with a partner/married	1.05 (0.87–1.26)	0.615		
Level of Education	Primary School	1.00 ref	--	1.00 ref	-
	Grade 12	1.13 (0.85–1.51)	0.404	1.23 (0.88–1.73)	0.231
	Diploma or Degree	1.25 (0.95–1.64)		1.13 (0.86–1.49)	0.391
	Postgraduate	0.63 (0.48–0.82)	0.001 *	0.61 (0.45–0.84)	0.003 *
Work in Hospital	Yes	1.00 ref	-	1.00 ref	-
	No	0.89 (0.82–0.97)	0.005 *	1.01 (0.72–1.41)	0.959
Years of experience	0–2	1.00 ref	-	1.00 ref	-
	3–6	0.75 (0.61–0.93)	0.007 *	0.67 (0.48–0.95)	0.023 *
	7–10	0.90 (0.80–1.03)		0.85 (0.61–1.18)	0.326
	More than 10	-	-	0.91 (0.73–1.13)	0.384
Facility location	Rural	1.00 ref	-	1.00 ref	-
	Urban	1.23 (0.92–1.63)		1.29 (0.98–1.70)	0.065
All forms of PPE Protection	Yes	1.00 ref	-		
	No	0.83 (0.52–1.34)	0.453		
Frequency of PPE Use	Frequently	1.00 ref	-	1.00 ref	-
	Always	1.81 (1.67–1.95)	0.000 *	-	-
PPE Training	Yes	1.00 ref	-	1.00 ref	-
	No	1.13 (1.04–1.23)	0.005 *	1.25 (1.00–1.58)	0.051 **
Vaccination	Yes	1.00 ref	-		
	No	0.93 (0.66–1.30)	0.669		
Comorbidity	Yes	1.00 ref	-		
	No	1.07 (0.77–1.50)	0.688		
Handling all Forms of Waste	Yes	1.00 ref	-		
	No	0.82 (0.59–1.16)	0.265		
Booster Shot	Yes	1.00 ref	-	1.00 ref	-
	No	1.13 (0.95–1.36)		0.90 (0.71–1.15)	0.407
Gloves and Mask	Yes	1.00 ref	-		
	No	1.21 (0.86–1.71)	0.265		
Frequency of	Once	1.00 ref	-	1.00 ref	-
Medical Waste	Three times	1.50 (1.00–2.25)	0.051 *	-	-
Collection	Four times	1.13 (0.60–2.09)	0.710		0.818
	Everyday	1.38 (0.89–2.00)		1.12 (0.88–1.44)	0.352
Availability Hand Washing Facility	Yes	1.00 ref	-	1.00 ref	-
	No	1.13 (1.04–1.24)	0.005 *	0.93 (0.73–1.19)	0.578

*—statistically significant; **—marginally statistical significant difference.

## Data Availability

Data are available upon reasonable request from the corresponding author.

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
