# Peer review of "A Pilot Study Exploring the Risk of SARS-CoV-2 Infection Among Employees Handling Healthcare Waste in Selected Healthcare Risk Waste Facilities in Johannesburg, South Africa"

_ijerph, 2025, doi:10.3390/ijerph22020243_

Round 1
Reviewer 1 Report
Comments and Suggestions for Authors
General comment
The paper reports a study of COVID-19 prevalence in employees working at healthcare waste disposal in South Africa. The topic is interesting to understand the risks. However, there are several aspects in which the paper is too qualitative and some details that it is better to discuss, see my specific comments. I suggest to consider the paper for publication after a revision step.
Specific comments
It is not clear why the mentioning of references starts from 18. The numbering usually is done in order of mention in the paper.
Introduction, please mention the recent work of Dinoi et al. (Science of the Total Environment 809, 151137 2022) and Chia et al (Nat. Commun., 11 (2020), p. 2800). That showed larger risks in healthcare facility for COVID-19 transmission.
It is mentioned often during COVID-19 pandemic. This is too vague please mention clearly what is the period of the study and stick to precise periods during discussion and presentation of results.
Table 2. What does it mean +ve/n%?
The prevalence of COVID-19 in the sample studied is not very relevant by itself. It should be compared to the prevalence in general population in the same period of the study and with prevalence of healthcare employees not dealing with waste.
Table 3. Please explain what is COR and AOR.
Lines 246-248. Again, this suggest to understand what is the differences compared to the average population and the other employees at risk in healthcare facilities.
Lines 265-270. It seems that the difference between males and females may be only related to the different distributions in the group studied. Males are the large majority.
Author Response
|
Comments 1: It is not clear why the mentioning of references starts from 18. The numbering usually is done in order of mention in the paper. |
|||||||||||||
|
Response 1: Numerical order was used Thank you for pointing this out. We agree with this comment. Therefore, we have ensured that reference in the text is implemented according to the journal requirements.
|
|||||||||||||
|
Comments 2: Introduction, please mention the recent work of Dinoi et al. (Science of the Total Environment 809, 151137 2022) and Chia et al (Nat. Commun., 11 (2020), p. 2800). That showed larger risks in healthcare facility for COVID-19 transmission. |
|||||||||||||
|
Response 2: Thank you for the comment. We have ensured that recent work from Dinoi and colleagues was included (line 67-69 and 270-272).
|
|||||||||||||
|
4. Response to Comments on the Quality of English Language |
|||||||||||||
|
Point 1: The quality of English does not limit my understanding of the research. |
|||||||||||||
|
Response 1: Thank you for the comment. |
|||||||||||||
|
Comments 1: It is not clear why the mentioning of references starts from 18. The numbering usually is done in order of mention in the paper. |
|||||||||||||
|
Response 1: Numerical order was used Thank you for pointing this out. We agree with this comment. Therefore, we have ensured that reference in the text is implemented according to the journal requirements.
|
|||||||||||||
|
Comments 2: Introduction, please mention the recent work of Dinoi et al. (Science of the Total Environment 809, 151137 2022) and Chia et al (Nat. Commun., 11 (2020), p. 2800). That showed larger risks in healthcare facility for COVID-19 transmission. |
|||||||||||||
|
Response 2: Thank you for the comment. We have ensured that recent work from Dinoi and colleagues was included (line 67-69 and 270-272).
|
|||||||||||||
|
4. Response to Comments on the Quality of English Language |
|||||||||||||
|
Point 1: The quality of English does not limit my understanding of the research. |
|||||||||||||
|
Response 1: Thank you for the comment. |
|||||||||||||
|
|

Reviewer 2 Report
Comments and Suggestions for Authors
1. The sample size in this study is too small to draw robust conclusions. How do the authors ensure that this small sample is representative of the larger population of medical waste handlers? Additional justification or discussion on the representativeness of the sample would strengthen the study's credibility. Emphasize this limitation more clearly in the conclusion and suggest directions for future, larger-scale studies.
2. The abstract provides a succinct overview but should include the main statistical findings (e.g., odds ratios) to underline the key results.
3. The introduction highlights the global and regional significance of the topic. However, it could better articulate the research gap being addressed. Include references to recent, similar studies for comparative context.
4. Mention if the structured questionnaire was pre-tested or validated before use.
5. The tables are informative, but some variables lack sufficient discussion in the text.
6. The lack of significant associations for certain factors like vaccination status should be discussed further. Could it reflect sample limitations or other biases?
7. The discussion appropriately references similar studies but should delve deeper into why certain results differ from or align with the literature.
8. Clearly outline next steps.
9. Several references are outdated or generic. Update with recent, region-specific studies where possible.
10. The tables are clear, but graphical representations might make the results more accessible.
11. Some sentences are overly long and could be rewritten for clarity.
No
Author Response
|
|
Comments 1: The sample size in this study is too small to draw robust conclusions. How do the authors ensure that this small sample is representative of the larger population of medical waste handlers? |
|
|
Response 1: Thank you for the comment, we agree with the reviewer that the sample size is not large enough to draw robust conclusion, as we have indicated in the conclusion and recommendation this study is a pilot study and paves a way for a study with a larger size. Greater detail in line 330-333.
Furthermore, the selection of HCW handlers to participate in the pilot study was done using random and convenience sampling methods, which selected multiple HCW facilities across Johannesburg.
Comment continuation 1.2: Additional justification or discussion on the representativeness of the sample would strengthen the study's credibility. Additional justification or discussion on the representativeness of the sample would strengthen the study's credibility. Response 1.2: Thank you for the comment. According to the South African Department of Forestry Fisheries and the Environment, there is a total of 11 registered HCW facilities in the Gauteng province, thus, 8 facilities were selected to participate in the study in Johannesburg. Thus, fairly representative of the study population. More details are provided from line 81 to 84.
Comment Continuation 1.3: Emphasize this limitation more clearly in the conclusion and suggest directions for future, larger-scale studies. Response 1.3: Thank you for the comment and the suggestion, the limitations are highlighted in page 10 (line 321-325) and direction for future studies is also highlighted: ‘’This pilot study presents a number of limitations, and bias, including selection bias during sampling of study participants and recall bias, where participants may not recall critical information related to factors investigated in this study. Although the cross-sectional study aided in establishing the magnitude of the association between the study variables. The limitations to its use include a temporal association, thus inability to establish causal factors’’.
|
|
|
Comments 2: The abstract provides a succinct overview but should include the main statistical findings (e.g., odds ratios) to underline the key results |
|
|
Response 2: Thank you for the input, key results summarizing the study population and factors associated with SARS-CoV-2 from the adjusted regression model are shown (Table 3, page 7).
|
|
|
Comments 3: The introduction highlights the global and regional significance of the topic. However, it could better articulate the research gap being addressed. Include references to recent, similar studies for comparative context. |
|
|
Response 3: We agree with your comment. We have, accordingly, revised the text to emphasize this point under introduction on page 1, the research gap is articulated: “The spike in HCW due to severe acute respiratory syndrome coronavirus 2 (SARS-CoV-2) presented significant risk for bio-hazardous exposure to the environment and the health of those handling the waste” – line 41 to 43.
|
|
|
Comments 4: Mention if the structured questionnaire was pre-tested or validated before use. |
|
|
Response 4: Thank you for the comment- yes, the questionnaire was piloted/ pre-tested before use, in 10 participants, HCW handlers - page 3, line 106-107. |
|
|
|
|
|
Comments 5: The tables are informative, but some variables lack sufficient discussion in the text. |
|
|
Response 5: Thank you for the comment, results of key variables and findings such as frequency of PPE use, vaccination, comorbidity, all waste form, gloves and masks have been discussed – page 10, paragraph 5, line 278-285.
|
|
Comments 6: The lack of significant associations for certain factors like vaccination status should be discussed further. Could it reflect sample limitations or other biases? |
|
|
Response 6: Thank you for the comment, as indicated in the manuscript, the small sample size resulted in variables such as frequency of PPE use, vaccination, comorbidity, all waste form, gloves and masks not being able to provide an output for the AOR, thus, insufficient observation – line 283-285. Bias is therefore discussed in detail in the paragraph before conclusion (line 321-322). |
|
|
|
Comments 7: The discussion appropriately references similar studies but should delve deeper into why certain results differ from or align with the literature. |
|
|
Response 7: Thank you for the comment, the discussion has been improved to include recent literature/ studies showing why our results differs from the one in literature an example is provided for comorbidities (line 288-294).
|
|
Comments 8: Clearly outline next steps. |
|
|
Response 8: Thank you for the comment, the next steps and way forward have been discussed as recommendations for future studies (line 330-333).
|
|
|
|
Comments 9: Several references are outdated or generic. Update with recent, region-specific studies where possible. |
|
|
Response 9: Thank you for the input. New updated references have been added - reference [13-17]. |
|
Comments 10: The tables are clear, but graphical representations might make the results more accessible. |
|
Response 10: Thank you, we agree with your comment. Figure 2 has been added to describe the cumulative incidence of SARS-CoV-2 (page 6).
|
|
Comments 11: Some sentences are overly long and could be rewritten for clarity. |
|
Response 11: We Agree. Furthermore, we have, accordingly, revised to improve the language and grammar of the manuscript. Moreover, to provide clarity. |
|
4. Response to Comments on the Quality of English Language |
|
Point 1: (x) The English could be improved to more clearly express the research. |
|
Response 1: Thank you for your comment. We have, accordingly, revised to improve the language and grammar of the manuscript. Moreover, to provide clarity. |
|
|

Reviewer 3 Report
Comments and Suggestions for Authors
A genuine pilot study of possible relevance, despite its very small dimension, about a relevant, often neglected public health subject; unfortunately, its presentaton at least is affected by significant faults in both terminology and presentation that condition the comprehensibility and the possibility of a reasonable appraisal of the results.
i) The terms "COVID-19" and "COVID-19 infection" are often used in such a way that the expression "SARS-CoV-2 infection" seems to be the correct one to be adopted in the context; see, for instance, the Lines 125, 138, and 141 and Table 1.
ii) Lines129-131: why "age, gender, education, years of experience and healthcare risk waste (HCRW) facility" have been considered as "confounders"?
iii) Line 169 and then later, particularly at Line 254: "the prevalence of COVID-19" is a certainly improper term (perhaps "cumulative incidence" is the correct one?).:
iv) Line 103: the term "population selected" has to be assumed as "target population"?
v) Lines 110-112: how many the "eligible partipants" were? how many out of "eligible" refused to answer? the possibility of a selection bias has been considered?
Other more specific observations:
- Lines 23-25 and then later: 7 out of 33 participants tested positive for "COVID-19" (see above), but the Authors did'nt give information about the timing of the positive tests;
- Line 29 and then later: the acronym "PPE" (reasonably "Personal Protection Equipment") has not been defined: it refers to FFP2 facial masks, medical masks, gloves, other...?
- Lines 60-61: "SARS-CoV-2" instead of "COVID-19";
- Lines 97-101: the study disegn (a cross-sectional one) is correct, but just its "advantages" and not its limitation have been considered; please, see the attached paper or something equivalent;
- Table 1: something has to be verified, for 32 records result for the variable "Marital Status", 30 for the variable "Level of Education", 32 for the variable "Facility Location", 31 for the variable "USE of PPE", 32 for the variable "Comorbidity", 32 for the variable "Vaccination Status".
- Lines 256-257: the meaning of the sentence "In this study, the years of experience (p = 0.50) were statistically significant with COVID-19 infection" is unclear.

Author Response
|
Comments 1: A genuine pilot study of possible relevance, despite its very small dimension, about a relevant, often neglected public health subject; unfortunately, its presentation at least is affected by significant faults in both terminology and presentation that condition the comprehensibility and the possibility of a reasonable appraisal of the results. |
|
Response 1: Thank you for the comment. The paper has been critically revised and edited for clear comprehension and cohesion of information to enhance understanding.
Comments 2: The terms "COVID-19" and "COVID-19 infection" are often used in such a way that the expression "SARS-CoV-2 infection" seems to be the correct one to be adopted in the context; see, for instance, the Lines 125, 138, and 141 and Table 1. |
|
Response 2: Thank you for pointing this out. COVID-19 and COVID-19 infection has been written throughout the paper as SARS –CoV-2 and SARS–CoV-2 infection. |
|
|
|
Comments 3: Lines129-131: why "age, gender, education, years of experience and healthcare risk waste (HCRW) facility" have been considered as "confounders"? Response 3: Thank you for the comment. The sentence has been removed, age, gender, education, years of experience and healthcare risk waste (HCRW) facility are possible risk factors for the outcome variable, SARS- CoV-2 infection and not confounders (which will be associated / a risk factor for both the outcome and the independent variable).
Comments 4: Line 169 and then later, particularly at Line 254: "the prevalence of COVID-19" is a certainly improper term (perhaps "cumulative incidence" is the correct one?).: |
|
Response 4: Thank you for noticing this. Prevalence has been replaced with cumulative incidence (line 75, 142-143 and 161).
|
|
Comments 5: Line 103: the term "population selected" has to be assumed as "target population"? |
|
Response 5: Thank you for the comment, however, has been removed during the process of the manuscript revisions. |
|
|
|
Comments 6: Lines 110-112: how many the "eligible partipants" were? how many out of "eligible" refused to answer? the possibility of a selection bias has been considered? |
|
Response 6: The data collection is described in the methods in detail. We have removed the word eligible as this may not be an appropriate word for this pilot study. The section has been revised as “The questionnaire was designed on Google Forms and was sent to participants who consented to participate in the study. Participants who were unable to complete the survey online were interviewed telephonically, while others were provided with hard copies of questionnaires’’. (line 102-105).
|
|
Comments 7: Lines 23-25 and then later: 7 out of 33 participants tested positive for "COVID-19" (see above), but the Authors did'nt give information about the timing of the positive tests; |
|
Response 7: Thank you, we agree with your comment. The SARS-CoV-2 test were those who recalled and were tested positive for COVID-19 during the period of the study (November 2021- Nov 2023) – line 70. |
|
|
|
Comments 8: Line 29 and then later: the acronym "PPE" (reasonably "Personal Protection Equipment") has not been defined: it refers to FFP2 facial masks, medical masks, gloves, other...? |
|
Response 8: Thank you, we agree with the comment. This has been revised and discussed in greater detail in the introduction (paragraph 2, line 61-64).
|
|
Comments 9: SARS-CoV-2" instead of "COVID-19" |
|
Response 9: Thank you for your comment. All text that previously indicated COVID-19 were revised to SARS-CoV-2 please see response 2 above.
Comments 10: Lines 97-101: the study disegn (a cross-sectional one) is correct, but just its "advantages" and not its limitation have been considered; please, see the attached paper or something equivalent; Response 10: We agree with your comment. The limitations of using a cross-sectional study design are highlighted in the discussion, please see page 10, paragraph 4, line 323-326. This includes the “temporal association, meaning the factors found significant in the model are not direct causal factors for SARS-CoV-2”.
|
|
Comments 11: Table 1: something has to be verified, for 32 records result for the variable "Marital Status", 30 for the variable "Level of Education", 32 for the variable "Facility Location", 31 for the variable "USE of PPE", 32 for the variable "Comorbidity", 32 for the variable "Vaccination Status" |
|
Response 11: Thank you for pointing that out. Table 1 has been updated, variables with missing data are shown and the percentages add to 100% and frequency add to 33, which is the total number of study participants – see page 4 and 5.
|
|
Comments 12: Lines 256-257: the meaning of the sentence "In this study, the years of experience (p = 0.50) were statistically significant with COVID-19 infection" is unclear. |
|
Response 12: Thank you for the correction. We have revised the sentence: “Among the factors analysed, years of experience showed a significant association with SARS-CoV-2 infection (p=0.050)” – line 152-154. |
|
|
|
4. Response to Comments on the Quality of English Language |
|
Point 1: (x) The quality of English does not limit my understanding of the research. |
|
Response 1: Thank you for your comment. |

Round 2
Reviewer 1 Report
Comments and Suggestions for Authors
Authors improved the paper in the first revision step and answered to my questions. I believe that it can be accepted for publication in the current form.
Author Response
Authors improved the paper in the first revision step and answered to my questions. I believe that it can be accepted for publication in the current form.
Thank you for the comment.

Reviewer 3 Report
Comments and Suggestions for Authors
The study design has been adequately described and the limitations of the research have been correctly exposed and discussed.
Just some further, minimal remarks.
Line 24 - "however these factorswere not significant": the meaning is unclear.
Line 30: "modifying" instead of "increasing" (some of the listed factors can actually lower, not increase the risk levels)
Line 31: "Nonetheless" is unnecessary
Line 58: "Kenya, and Nigeria" instead of "Kenya, Nigeria, and"?
Line 124; "having a positive laboratory test result" instead of "having a laboratory test result"?
Table 2 - row "Use of PPE": unclear data
Line 215: "a role in the risk of infection" instead of "a role in infection"?
Line 217 "during reported"?
Author Response
Comments 1: Line 24 - "however these factorswere not significant": the meaning is unclear.
Response 1: Thank you for the comment, this has been revised as “however, these factors were not significant as risks for SARS-CoV-2” (line 24 and 25).
Comments 2: Line 30: "modifying" instead of "increasing" (some of the listed factors can actually lower, not increase the risk levels)
Response 2: Thank you for the input, this has been revised (line 30).
Comments 3: Line 31: "Nonetheless" is unnecessary
Response 3: We agree with your comment. We have, accordingly, revised the text (line 31)
Comments 4: Line 58: "Kenya, and Nigeria" instead of "Kenya, Nigeria, and"?
Response 4: Thank you for the comment, we have revised this (line 58).
Comments 5: Line 124; "having a positive laboratory test result" instead of "having a laboratory test result"?
Response 5: Thank you for the comment, we have revised this (line 124).
Comments 6: Table 2 - row "Use of PPE": unclear data
Response 6: Thank you for the comment, however, as indicated in the table, there were only 31 respondents who answered (yes), none who answered (no) and 2 who accounted for missing data. The frequencies, therefore, make up 100%.
Comments 7: Line 215: "a role in the risk of infection" instead of "a role in infection"?
Response 7: Response 8: Thank you for the input, this has been revised (line 214).
Comments 8: Line 217 "during reported"?
Response 8: Thank you for the comment, this was a grammatical error, but it has been eliminated (line 216).
4. Response to Comments on the Quality of English Language
Point 1: (x) The English is fine and does not require any improvement.
Response 1: Thank you for your comment.
